# A Systematic Review of the Development and Psychometric Properties of Loneliness Measures for Children and Adolescents

**DOI:** 10.3390/ijerph18063285

**Published:** 2021-03-22

**Authors:** Aimée Cole, Caroline Bond, Pamela Qualter, Marlies Maes

**Affiliations:** 1Salford Educational Psychology Service, Burrows House, 10, Priestley Road, Worsley, Salford M28 2LY, UK; 2Manchester Institute of Education, University of Manchester, Manchester M13 9PL, UK; caroline.bond@manchester.ac.uk (C.B.); pamela.qualter@manchester.ac.uk (P.Q.); 3School Psychology and Development in Context, KU Leuven, Tiensestraat 102, 3000 Leuven, Belgium; marlies.maes@kuleuven.be; 4Research Foundation–Flanders (FWO), Egmontstraat 5, 1000 Brussels, Belgium; 5Interdisciplinary Social Science, Utrecht University, Padualaan 14, 3584 CH Utrecht, The Netherlands

**Keywords:** loneliness, measurement, childhood, adolescence, psychometrics

## Abstract

This paper reviews the three most commonly used measures of loneliness for children and adolescents (children: Loneliness and Aloneness Scale for Children and Adolescents [LACA] and Children’s Loneliness and Social Dissatisfaction Scale [CLS]; adolescents: UCLA Loneliness Scale [UCLA] and LACA). Loneliness is a pertinent issue across populations and affects the mental health and academic achievement of children and adolescents. To date, there has been no thorough examination of the loneliness measures for this age group. We examine how each of the three measures was developed, and assess the psychometric properties of those measures, gaining insight into whether they are valid and reliable assessments of loneliness. Results suggest that the UCLA Loneliness Scale is the most popular measure of loneliness for use with adolescents, but it does not have robust psychometric properties for that group. For children, the CLS appears most suitable. Results of the review identify gaps in aspects of measure development, with no measure having been developed with children or adolescents. Implications for future loneliness measurement research are considered.

## 1. Introduction

Loneliness is a painful experience, associated with feeling unhappy, unloved, restless, and generally despondent across different age groups, including school-aged children [1]. Aligned with the most popular conceptualization [2], loneliness is experienced when one perceives a discrepancy between actual and desired social relationships, and this discrepancy can be experienced in relation to either or both the quantity and quality of one’s relationships. Loneliness has been associated with the absence of play partners and negative relationships in childhood, and a lack of close friends and peer rejection during adolescence [3]. In addition, school-based victimization has been found to be associated with loneliness during adolescence [1] and young adulthood [4], suggesting that loneliness can also ensue from negative social relationships. 

Loneliness has been related to a host of negative outcomes, including worse academic attainment, emotional health difficulties and sleep quality in youth [3,5,6,7]. Whilst our understanding about the negative effects of loneliness is increasing, to date there has been no review of the assessment of loneliness for children and adolescents. Such a review is particularly important given the current COVID-19 pandemic and national and regional lockdowns that children are experiencing. The closing of schools associated with the COVID-19 pandemic has raised concerns about increasing loneliness among youth, given their absence from friends and a peer social structure; there is a need to use appropriate measurement to identify whether that is the case. Loneliness in children and adolescents is commonly measured using specific assessments, that is, UCLA Loneliness Scale (UCLA) [8] Children’s Loneliness and Social Dissatisfaction Scale (CLS) [9], and Loneliness and Aloneness Scale for Children and Adolescents (LACA) [10]. However, a systematic overview of development procedures and psychometric properties of those measures with youth is not yet available. 

Currently, loneliness among youth is not screened in the way that depression or internalizing problems are, whereby individuals complete self-report measures with pre-determined cut-offs indicating difficulties [11]. However, there are benefits of screening, including identifying those in need of extra support, and prevention of the concurrent and prolonged mental health problems in youth that are linked to loneliness. To do this effectively, there must be robust measurements available. While not initially developed primarily for screening, there are three measures of loneliness that are commonly used to assess loneliness among children and adolescents. However, there has been no systematic review examining their reliability and validity, nor any discussion about their development, including whether they followed guidelines to create robust and useful measures. 

### Developing Measures

Measure development literature recommends inductive and deductive methods during item creation, to limit contamination and support valid depiction of relations to other constructs [12,13]. The three main steps of measure development are (1) specifying observable characteristics, (2) determining the extent to which they measure the same thing using empirical research and statistical analysis, and (3) performing experiments to determine the extent to which measures are consistent with established views of the construct [14]. Qualitative data from target populations that outline opinions and experiences of the construct are also required to inform understanding of the subjective experience of the concept [15,16]. Valid and reliable measurement is scientifically fundamental and essential for robust research [17,18] and replicability. Thus, in the current study, we (1) explore how the loneliness measures used to collect data from children and adolescents were developed, and, indeed, whether they followed the steps for successful measurement development, and (2) explore the reliability and validity estimates of each measure as they are presented in papers that have subsequently explored their psychometric properties. 

## 2. Materials and Methods

### 2.1. Literature Search Strategy and Review Process

The MASLO database, details of which are described elsewhere [19], includes studies that apply one of the seven most used loneliness measures for different ages, published between 1978 and 2013. Literature searches were conducted in 2013, yielding 3658 results, of which 1585 were excluded due to not including one of the seven loneliness measures, written in languages other than Dutch, English, French or German, or were irretrievable. Subsequently, papers were read in detail, with further exclusions made in the absence of detail regarding methodology, or the absence of numerical information. For the current study, the MASLO database [19] was screened for papers focused on reliability and validity testing of those measures among youth, with the Loneliness and Aloneness Scale for Children and Adolescents (LACA) [10], UCLA Loneliness Scale (UCLA) [8] and Children’s Loneliness and Social Dissatisfaction Scale (CLS) [9], being drawn from the database, including 1821 papers in total. Between September 2019 and March 2020, additional searches for articles published between 2014 and 2020 in Scopus, PubMed, and PsychInfo were conducted. Key search terms combined included “loneliness scale for children and adolescents”, “loneliness measure”, “reliability”, and “validity”. Additional searches, using the titles of each of the measures, were also conducted, yielding an additional 2345 results. Figure 1 outlines the review process.

After removing duplicates and screening for papers not measuring reliability and validity of one of the three loneliness measures, 64 papers, drawn from both the MASLO database and new database searches, were screened against the inclusion criteria: (1) the study explicitly tests the reliability and validity of the loneliness measure, (2) sample participants under 18 years old (3) a Cronbach’s alpha for the loneliness (sub)scale was presented, and (4) paper was written in English. Five papers for the LACA, nine for the UCLA, and six papers for the CLS, were included in the final review. Development papers for each measure, with the addition of a pre-development paper for the LACA (see Table 1), were included in the review database, subsequent searches for such were conducted using the combined search terms “development”, “UCLA”, “LACA”, “CLS”, “loneliness”, and “scale”. An additional five studies, known to the researchers, were examined against the inclusion criteria, with two being included. 

Through screening procedures, reference to “pure” loneliness measure adaptation, resulted in further searches (using the terms “pure”, “loneliness”, “CLS” and “measure”) and the inclusion of four papers. Additionally, upon reading the LACA development paper, Marcoen and Brumagne’s (1985) [20] paper was cited as the research paper from which items were drawn, leading to its inclusion in the current review. 

**Figure 1 ijerph-18-03285-f001:**
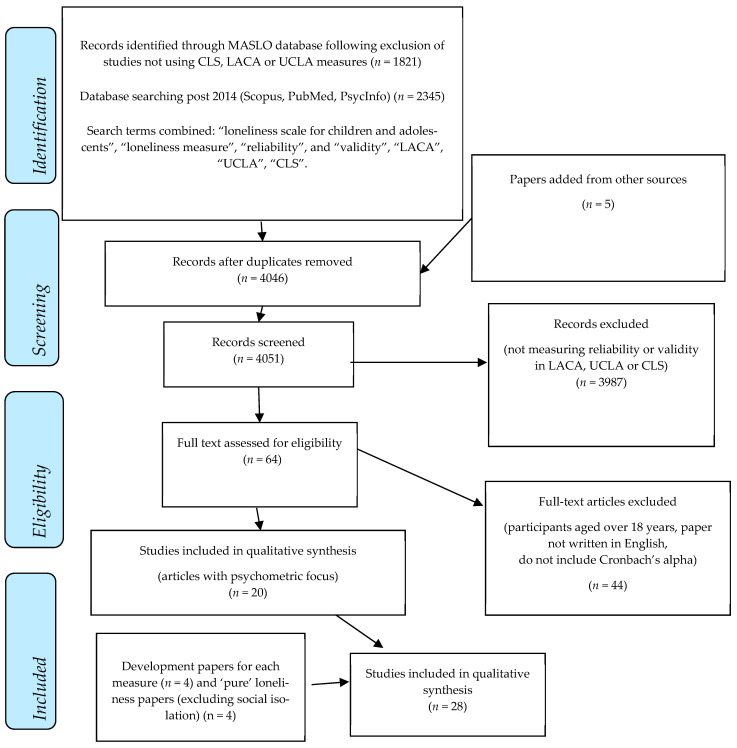
PRISMA diagram [21].

### 2.2. Data Classification

Data classification is discussed first in relation to the development papers, where we used the critical appraisal tool. Then, we review the psychometric properties of each of the loneliness measures, exploring the inclusion of children and adolescent voices in the development of the measures and the reliability and validity as documented in subsequent studies. 

### 2.3. Development of a Critical Appraisal Checklist

The quality and suitability of development processes for each of the measures were determined using a critical appraisal tool, created by the researchers from existing well-established checklists (details of checklist items can be found in Appendix A). No pre-existing frameworks captured the aims of the current research, so measure development research was consulted to support the inclusion criteria. The tool was informed by four quantitative evaluation frameworks including COSMIN [22], JBI Appraisal Checklist for Studies Reporting Prevalence Data [23], Evidence Based Medicine and Practice checklist [24], and the University of Manchester quantitative evaluation research checklist [25]. Checklist drafts were trialed to refine each element and establish clear wording and weighting of statements (completed checklists and explanations of items can be found in the Appendix A; completed critical appraisal checklists for each measure can be found in Appendix A). Each development paper, plus the Marcoen and Brumagne (1985) [20] paper, was read in full by the first and third author, assessing 13 “core” expectations of measure development, and two supplementary statements for the inclusion of a factor analysis or structural equation modelling, and invariance testing in subsequent papers. The papers selected were the first papers for each measure, outlining development procedures. During this process, researchers looked at quality appraisal literature to support the use of qualitative quality decisions, using overall inter-rater agreements of “high”, “medium”, or “low” [24,26]. Reviewers awarded two stars when the quality statement was met “to a great degree”, one star where it was “partially met”, and a hyphen where the information was “incomplete or omitted”. Moderation discussions ensured consistent interpretation and application of the checklist. Papers with between one and four two-star ratings were deemed “low quality”. Those with between five and eight two-star ratings were noted as “medium quality”; nine or more two-star ratings were deemed “high” quality papers (see Table 1). 

### 2.4. Psychometrics

Following quality appraisal of development procedures, we examined the psychometric properties, focusing on the reliability and validity of each of the measures, in subsequent papers (see Table 2, Table 3 and Table 4). Papers employing an increasingly used brief version of the CLS to represent ‘pure’) loneliness items were also included (items from CLS adapted by Ladd et al. (1996) to represent loneliness separate from social isolation). To further understand how the measures have been used since development we also examined measurement adaptations and sample characteristics in subsequent papers. 

### 2.5. Reliability and Validity 

In the quality framework, reliability and validity are relevant and representative of the loneliness construct. Cronbach’s alpha values of 0.8 and above were considered acceptable levels of internal consistency both in development papers and subsequent research [27]. Content- and criterion-related validity were considered through reference to the inclusion of interviews with children and adolescents [28], explanations of theory underpinning the measures, concept definitions [29], and reference to existing constructs of loneliness in the development papers. 

## 3. Results

### 3.1. Scale Overview

The LACA extends the Louvain Loneliness Scale for Children and Adolescents (LLCA) [20] subscales (peer and parental loneliness), through the incorporation of two new subscales measuring positive and negative attitudes to aloneness and social isolation. The original development paper describes a 48-item measure, with four integrated subscales, which was tested on a sample of 444 children and adolescents from grade 5 to 11 (aged 10–16 years), the widest age range across the development papers of the three measures. Subscales relate to loneliness in parental relations (L-PART), loneliness in peer relations (L-PEER), affinity for aloneness (A-POS), and aversion to aloneness (A-NEG). Subscales were not revised following testing.

The UCLA, based on Sisenwein’s (1964) [30] scale, was developed and tested with young adults [8] using a 5-point scale. Researchers drew 25 items from a 75-item pool, excluding “very extreme statements”. The scale was revised following analysis, leaving a final 20-item measure, such that a revised scale correlation of items with the total loneliness scores and internal consistency, was assessed. Additionally, concurrent validity was explored through correlation with self-reports about current loneliness, comparison between scale scores across the lonely and comparison sample and self-ratings of feelings associated with loneliness. The UCLA is often used with older adolescents, and less often children, though some items are suggested for use with children by the Office of National Statistics (ONS) to measure childhood loneliness [31].

The CLS [9] was developed to explore loneliness and social dissatisfaction. The 24-item scale comprises 16 target and eight “filler” items, not included in the final loneliness score. Its original psychometric study included a sample of 522 children between grades three and six (aged 8–12 years). After two weeks, a sociometric measurement was administered to explore whether classroom peer group status was directly related to loneliness. The scale did not undergo post-test alterations.

### 3.2. Quality Appraisal

The CLS provided the clearest definition of loneliness and research questions, referring to feelings of loneliness and social dissatisfaction research with children, and aimed to develop a reliable measure of loneliness. The UCLA development outline provided little information regarding overarching constructs, or justification for measure development, referring only to the “lack of a simple and reliable method of assessment” [8] (p. 290). The LACA developers highlighted “age-linked feelings of loneliness” [10] (p. 1025) in youth and made distinctions regarding emotional and social loneliness. Target populations were largely well defined, although they varied between children (CLS) [9] undergraduate psychology students (UCLA) [8], and late childhood and adolescence (LACA) [10]. Furthermore, the ages of those in the development samples ranged between third grade (CLS) and “young adults” (UCLA), suggesting that development samples were not representative of subsequent populations in which the measures are used. Theoretical underpinnings of the measures varied: the LACA was rated most highly, describing the need to “cover related constructs of positively and negatively experienced aloneness” [10] (p. 562). 

Drawing on measure development best practice guidance, no development paper was awarded a score for core expectations related to content validity and none interviewed children and adolescents when developing their conceptualization of loneliness. UCLA items were drawn from “20 psychologists describing the experience of loneliness” [8] (p. 291), and statements from Eddy’s (1961) [32] measure, and omitted replication details. LACA items were drawn from Marcoen and Brumagne’s (1985) [20] “original scale”, but did not describe items; and the CLS paper did not outline any item inclusion strategies. 

Appropriate data analysis processes (Appendix A) were defined, including correlations between loneliness scale scores and relevant related constructs, and a Cronbach’s alpha calculated with only the UCLA omitting factor analyses. All measures presented Cronbach’s alphas above 0.8. 

Regarding cross-cultural validity, the UCLA development paper examined the effects of region and sex. Each measure, to some extent, compared scores with suitable and related variables. The LACA examined how age, sex, parental occupation, social integration, home environment, ecological situation (hometown size and home conditions), and psychological factors affected understanding and response to items. The CLS compared scores to sociometric status, examining links with friendship nomination, a relation deemed modest by researchers, due to the suggestion loneliness most closely links with perceptions of friendship rather than individual experience. Despite this, none of the measures explicitly explored cross-cultural validity. 

### 3.3. Quantitative Synthesis Psychometrics

Subsequent studies implementing the LACA, UCLA, and CLS with CA were reviewed with a specific focus on exploring the reliability and validity of each measure (see Table 2, Table 3 and Table 4).

#### Reliability and Validity 

Subsequent psychometric studies using the LACA included samples of children and adolescents between grades 5 and 12 (ages 10–18 years), conducted between 1987 and 2020. Each used 4-part response category scales, and either 36 or 48 items; those studies that used the 36 items dropped the L-PART (parent loneliness) subscale. The UCLA, the most widely adapted measure in subsequent studies, least often used with children, included 20, 8-, 6- and 4-item adaptations in psychometric studies between 1978 and 2020, with children and adolescents aged between 11 and 18 years old. Studies with the CLS often used younger samples, between 5 and 13 years old. In its original 24-item form, between three and five response categories were commonly used, prior to extraction of “pure” items [53]. Four papers implementing “pure” loneliness items with participants aged between five and 11 years, presented inconsistent item numbers, and largely insufficient Cronbach’s alphas, ranging from 0.75 to 0.87 [52,53,54,55]. Papers included between three and five response categories, with one paper omitting that detail [53]. Although broadly similar, “pure” loneliness items were not consistent across papers, with both 5-item measures referring to “three items relating to loneliness and an additional two semantically related items”, without clear explanation [53,54].

A Cronbach’s alpha of 0.8 or above was considered an acceptable rating of internal consistency, although a higher alpha does not directly illustrate greater internal consistency, as, if an alpha is too high, perhaps some items are measuring the same thing, in a different form. The largest discrepancy in alphas using the LACA was in the parental relationship subscale (0.81 and 0.93), suggesting difficulties in defining this type of loneliness. The original 20-item UCLA appears the most internally consistent version (between 0.71 and 0.96), although an alpha of 0.96 could suggest some unnecessary items. Contrarily, the lowest alpha was presented for the 4-item UCLA measure (0.31) [41] suggesting decreased reliability following item reduction. Lower alphas were further demonstrated through sample comparisons with populations deviating from the development sample (Hispanic ethnicity and those diagnosed with Attention Deficit Hyperactivity Disorder (ADHD), providing evidence of reduced reliability with populations other than young adults at university, who the scale was developed for. With the CLS, lower Cronbach’s alphas (0.79/0.76) were present in the studies using younger participants, highlighting potential problems with the measure often used with early-primary school aged children (youngest 5.3 years). 

Both the CLS and UCLA development papers presented higher internal consistency estimates compared with subsequent studies. That raises questions about their generalizability across diverse child and adolescent samples. Studies with “pure” loneliness items were included in the review and comprised samples of children aged between 5 and 11 years. Only one demonstrated an acceptable Cronbach’s alpha value with children aged between 9 and 11 years (0.87) [55]. 

Omitted data make mean score comparisons inconsistent, although the UCLA study presented equal scores for male and female samples, 40.34 (7.62), using the 20-item measure [41]. Mean and SD variations for age and nationality subgroups are demonstrated in the CLS. Goossens and Beyers (2002) [33] demonstrated intercorrelations/concurrent validity between the LACA peer-related loneliness subscale and the CLS, although none of the development papers explicitly explored the effects of culture or measurement invariance upon measure completion. 

## 4. Discussion

The current review examined development processes and subsequent reliability and validity testing of the three most widely used loneliness measures for youth: the CLS, UCLA and LACA. These measures were developed some time ago and when considered in relation to recent standards for measure development, it is clear that each omitted key processes, including interviews with children and adolescents, exploration of population variance, and comparisons with suitably related variables. Subsequent studies present scale adaptations with reduced reliability, and there is once more minimal investigation of the effects of culture. Absent also is exploration of concurrent validity. Inconsistencies are evident in research outlines, including descriptions of sample ages, response options, and item selection. In a time of increased focus and understanding of the impact of loneliness on youth, an appropriate measure for exploring the contemporary experience of loneliness among youth is required.

Despite strengths in psychometric development across the three measures, qualitative exploration of loneliness experiences with target populations is absent [15,56]. None of the measures were developed from interviews with youth, suggesting that their views of the loneliness experience did not inform the measures. Partial support for four latent constructs of loneliness (peer-related loneliness, family-related loneliness, aversion to being alone and affinity for being alone), across the LACA and CLS has been suggested, with social loneliness best measured by the CLS and the peer-related subscale of the LACA [33]. Prior to generation of these now well-established scales, it was perhaps difficult to accurately conclude their concurrent validity. However, the current literature review suggests that development procedures were incomplete, and subsequent use of scale items and response categories has been inconsistent. Therefore, the requirement for interviews conducted with youth in order to increase understanding and provide a foundation for establishing concurrent validity of the scales is highlighted. 

“Pure” loneliness measures (of the CLS) sought to further extract loneliness from close constructs to support specific intervention. However, that measure is narrowly explored and, to date, inconsistently administered [52,53,55].

Development samples differed from target populations for the measures, potentially reducing validity and reliability [57], as score comparisons are not with demographically similar individuals. If the age range of respondents in subsequent papers is expanded, construct validity is questionable because loneliness is experienced differently across development [3]. The lack of cross-cultural perspectives [58] significantly undermines the generalizability of the measures. These issues present challenges for those seeking to explore loneliness in diverse groups of children and adolescents. Future research should consider the possible impact of virtual interactions and friendship upon feelings of loneliness, and subsequently during measure completion and item understanding, particularly following the recent impact on youth mental health following the COVID-19 pandemic [59]. Researchers found variation in aspects of validity considered across the measures, and suggest further consideration of diverse development samples, matching the age of proposed audience for the measure, is required, along with consideration of cross-cultural validity and measurement invariance to support the use of these measures with present youth populations.

Since development, scale adaptations have been subject to inconsistent reliability and validity testing, with varied use of items and category responses. Test users may choose loneliness measurements by generalizing internal consistency coefficients from original scales, but, if concept development is flawed, then adapted scales have issues of inaccuracy because they are based on inaccurate concepts. Three items taken from the UCLA are currently recommended by the ONS and UK government [31], as the best measure of loneliness in youth following item revision and qualitative testing for ease and interpretation with young people aged 10 to 15 years. However, the current review has highlighted this was not the intended audience, and exploration of reliability and validity was absent, and so encouraging wider use of a similarly adapted scale, perhaps also in other countries, is founded on incomplete evidence. 

The “pure” loneliness subscale of the CLS, distinct from social dissatisfaction, demonstrated confusion and inconsistent item selection, with a lack of detail being unsupportive of replication [38,60]. The current review has highlighted, that the CLS appears most widely used with younger children. Practitioners keen to explore loneliness in youth should combine quantitative measures of loneliness with qualitative tools and knowledge of individuals to support a holistic picture and understanding grounded in the conceptualization of loneliness in youth. Consideration of age, which scale version is most suitable and the resulting psychometrics, along with the subsequent interventions that may be selected because of loneliness scores, is also pertinent for practitioners. 

### Limitations and Future Directions

The current review used robust review processes such as inter-rater agreement and collaborative development of the critical appraisal checklist. However, this tool was developed specifically for this project and further work to refine tools for the evaluation of development measures is warranted, including further development of evaluation tools that consider a range of aspects of validity required for robust measures. Our findings also identified the need for an updated approach to measuring loneliness in youth, one that addresses the key steps in measure development. Incorporating the views of children and adolescents and a more careful consideration of the effects of age and culture on how items are understood, and how loneliness is conceptualized, are particularly important to consider. Exploration of cut-offs is also needed if any measure is to be used for screening purposes. School practitioners should exercise caution when choosing a suitable tool, mindful of the highlighted issues during development and subsequent adaptations. To build on quantitative measures of loneliness, further exploration of youth understanding and views of loneliness, supported by well-established measures, could support adaptation of such, bringing them in line with contemporary conceptualizations of loneliness, suitable for diverse samples. 

## 5. Conclusions

The LACA, UCLA, and CLS, were insufficiently developed for use with children and adolescents, with additional gaps in our understanding of responses across diverse populations. High-quality, robust measures of loneliness are required with clear concept constructs, grounded in qualitative exploration of youth loneliness experiences. Further exploration of reliability, validity, and generalizability of the measures for different populations is required, to support intervention evaluation and screening uses of the measures. 

## Figures and Tables

**Table 1 ijerph-18-03285-t001:** Critical appraisal checklist ratings for each development paper.

Critical Appraisal Checklist Items		UCLA	CLS	LACA	
		Russell, Peplau and Ferguson (1978)	Asher, Hymel and Renshaw (1984)	Marcoen, Goossens and Caes (1987)	Marcoen and Brumagne (1987)
Core development procedure	Construct definition	**-**	******	*****	**-**
	Research questions outlined	*****	******	******	**-**
	Clear description of target population	**-**	******	******	******
	Theory outlined and described	**-**	*****	******	*****
	Interviews conducted with children and/or adolescents	**-**	**-**	**-**	**-**
	Appropriate qualitative data collection method for item identification	**-**	**-**	**-**	**-**
	Replication details included	*****	**-**	**-**	**-**
	Appropriate data analysis	******	******	******	******
Content validity/Internal structure	Interviews with experts regarding concept definition	**-**	**-**	**-**	**-**
	FA/structural equations model at development stage	**-**	******	******	******
Internal consistency	Cronbach’s alpha above 0.8	******	******	******	******
	Invariance testing	**-**	**-**	**-**	**-**
Cross-cultural validity/measurement invariance	Consideration of variance across different groups	*****	**-**	*****	*****
Responsiveness (comparison to gold standard)	Scores compared with related variables	*****	*****	*****	*****
	Suitable comparisons made	*****	*****	*****	*****
Overall quality decision	Number of ** ratings	2	6	6	4
				Mean: 5	
	Qualitative descriptor of overall quality	Low	Medium	Medium	

Core expectation; supplementary expectation; 2 stars (**) indicates that this was done well or in detail, 1 star (*) indicates that this was done partially, hyphen (-) indicates unclear or incomplete processes; overall quality: 1–4 ** ratings = low-quality paper; 5–8 ** ratings = medium-quality paper; 10–13 ** ratings = high-quality paper.

**Table 2 ijerph-18-03285-t002:** Subsequent research papers investigating the psychometric properties of the LACA.

Authors	Year	Title	Participant Age	Number of Participants	Number of Items	Response Categories	Language of Sample	Cronbach’s Alpha 1 = (L-PART) 2 = (L-PEER) 3 = (A-POS) 4 = (A-NEG)				Mean (Standard Deviation) 1 = (L-PART) 2 = (L-PEER) 3 = (A-POS) 4 = (A-NEG)						Correlations between Subscale Scores across Waves of Data Collection ¥ 1 = (L-PART) 2 = (L-PEER) 3 = (A-POS) 4 = (A-NEG)	
Development Paper: Marcoen and Brumagne [20]	1985	Loneliness among children and young adolescents	Grades 5 and 9	251	28	*η*	Dutch	Peer -related		Parent-related		*δ*						*δ*	
								0.88		0.68									
Development Paper: Marcoen, Goossens and Caes [10]	1987	Loneliness in pre-through late adolescence: exploring the contributions of a multidimensional approach	Grades 5–11 11–17 years	444	48	*η*	Dutch	1	2	3	4	1		2		3	4		
								0.88	0.87	0.80	0.81	18.80 (5.58)		21.08 (6.73)		29.70 (5.96)	30.94 (6.38)	*δ*	
Goossens and Beyers [33]	2002	Comparing measures of childhood loneliness: internal consistency and confirmatory factor analysis	Grades 5–6; Grade 5 mean age 10.5; Grade 6; mean age 11.5	292	48	*η*	Dutch	1	2	3	4	1		2		3	4		
								0.81	0.86	0.79	0.74	17.87 (5.04)		23.32 (6.83)		30.50 (6.13)	33.53 (5.79)	*δ*	
Maes, Van den Noortgate, and Goossens [34]	2015	A reliability generalization study for a multidimensional loneliness scale: the loneliness and aloneness scale for children and adolescents	79 studies	Elementary school (children) and secondary school students (adolescents)	*δ*	*δ*	Dutch, Arabic, Chinese, English, Greek, Hebrew, Italian, Spanish, Portuguese	1	2	3	4	1		2		3	4		
								0.86	0.87	0.79	0.80	1.65 (0.27)		1.80 (0.17)		2.64 (0.14)	2.58 (0.20)	*δ*	
Maes, Wang, Van den Noortgate, and Goossens [35]	2016	Loneliness and attitudes toward being alone in belgian and chinese adolescents: examining measurement invariance	Ages 11 to 15; Belgian mean age 12.80; Chinese mean age = 13.62	Belgian: 229 Chinese: 200	36	*η*	Sample 1: Dutch Sample 2: Chinese	Sample 1											
								1	2	3	4	1		2		3	4		
								*δ*	0.91	0.87	0.79	*δ*		21.39 (7.50)		29.06 (7.37)	32.04 (6.15)	*δ*	
								Sample 2											
								1	2	3	4	1		2		3	4		
								*δ*	0.89	0.83	0.87	*δ*		24.05 (7.19)		32.52 (6.53)	29.92 (7.17)	*δ*	
Danneel, Maes, Vanhalst, Bijttebier and Goossens [36]	2018	Developmental changes in loneliness and attitudes toward aloneness in adolescence.	Grades 9–10; Sample 1 = mean age 14.84; Sample 2 = mean age 14.82	Sample 1 = 834 Sample 2 = 968	48	*η*	Sample 1: Dutch Sample 2: Dutch	Sample 1											
								1	2	3	4	1		2		3	4		
								0.91–0.92*θ*	0.88–0.90	0.86–0.88	0.82–0.83	21.2 (6.96)	19.32 (6.48)		28.92 (6.36)		29.5 (5.88)	1	0.67–0.75
																		2	0.53–0.64
																		3	0.54–0.64
																		4	0.58–0.63
								Sample 2											
								1	2	3	4	1		2		3	4		
								0.90–0.93 *θ*	0.86–0.91	0.83–0.88	0.78–0.85	20.4 (6.48)		18.48 (5.52)		28.80 (5.88)	29.40 (5.28)	1	0.59–0.78
																		2	0.43–0.67
																		3	0.48–0.70
																		4	0.44–0.70
Danneel, Maes, Vanhalst, Bijttebier, and Goossens [37]	2018	Loneliness and attitudes toward aloneness in belgian adolescents: measurement invariance across language, age, and gender groups	Grades 7–12; Mean age Grade 7= 11.95 years, Mean age Grade 12= 17.16 French speaking mean age = 14.35 Dutch speaking mean age = 14.36	Dutch speaking: 641 French speaking: 641	48	η	Sample 1: Dutch Sample 2: French	Sample 1											
								1	2	3	4	1		2		3	4		
								0.91	0.90	0.85	0.80	20.53 (6.80)		21.17 (7.11)		31.15 (6.32)	31.72 (5.81)	*δ*	
								Sample 2											
								1	2	3	4	1		2		3	4		
								0.86	0.83	0.80	0.83	18.70 (5.99)		19.04 (6.63)		29.77 (6.66)	29.00 (6.25)	*δ*	

δ: no data present in paper; θ: across waves of data collection; η: response categories often (4) sometimes, seldom, never (1); ¥: ranges of 1 year stability correlations across three and four measurement waves in sample 1 and sample 2, respectively.

**Table 3 ijerph-18-03285-t003:** Subsequent research papers investigating the psychometric properties of the UCLA Loneliness Scale.

Authors	Year	Title	Participant Age	Number of Participants	Number of Items	Response Categories	Language of Sample	Cronbach’s Alpha			Mean (Standard Deviation)	
Development Paper: Russell, Peplau, and Ferguson [8]	1978	Developing a measure of Loneliness	Young adults	239	20	4 ***δ***	English	0.96			UCLA sample Males = 38.7 (11.0) Females = 40.2 (12.4) Tulsa sample Males = 38.6 (9.4) Females = 37.8 (9.7)	
Russell, Peplau and Cutrona[38]	1980	The revised UCLA loneliness scale: concurrent and discriminant validity evidence	Sample 1 = University students in first year Sample 2 = College students	Sample 1 = 162 Sample 2 = 237	20 + 19 Sample 1 = new items, 20 items made up scale Sample 2 = 20 + 10 positively worded items	4	English	Sample 1 = 0.94 Sample 2 = 0.94			Sample 2 = Males = 37.06 (10.91) Females = 36.06 (10.11)	
Mahon and Yarcheski [39]	1990	The dimensionality of the UCLA loneliness scale in early adolescents	12–14 years	326	20	4	English	0.84			β	
Neto[40]	1992	Loneliness among portuguese adolescents	14–17 years	217	6	4 δ	Portuguese	0.82			32.2 (7.0)	
Wilson, Cutts, Lees, Mapungwana, and Maunganidze [41]	1992	Psychometric properties of the Revised UCLA Loneliness Scale and two short form measures of loneliness in Zimbabwe	Mean age = 17.53	1354	20	4	English		Female	Male	Female	Male
								UCLA-20	0.72	0.71	40.34 (7.62)	40.34 (7.62)
								UCLA-8	0.60	0.56	17.67 (4.25)	17.08 (4.02)
								UCLA-4	0.38	0.31	8.27 (2.30)	8.14 (2.20)
Higbee and Roberts [42]	1994	Reliability and validity of a brief measure of loneliness with anglo-american and mexican american adolescents.	11–14 years	2614	8	4 δ	English	Anglo-American sample = 0.90 Hispanic sample = 0.87			7.13 (5.77)	
Russell [43]	1996	UCLA loneliness scale (version 3): Reliability, validity, and factor structure	English	489 students (part of a larger sample including 310 nurses, 316 teachers, 301 elderly)	20		English	0.92			40.08 (9.50)	
Neto [44]	2001	A short-form measure of loneliness among second-generation migrants.	15–18 years	109	6	4	Portuguese	0.71			β	
Lasgaard [45]	2007	Reliability and validity of the danish version of the UCLA loneliness scale.	13–16 years	224	20	4 1 = never 4 = always	English	Adolescents with ADHD = 0.84 Sample from regular schools = 0.91			Adolescents with ADHD = 37.6 (7.94) Sample from regular schools = 37.69 (10.23)	
Goossens, Klimstra, Luyckx, Vanhalst, and Teppers [46]	2014	Reliability and validity of the Roberts UCLA Scale (RULS-8) with dutch-speaking adolescents in belgium	12–18 years Sample 1 = grades 7–8, 12 and 13 years Sample 2 = grades 9–12, 14–18 years Sample 3 = grades 7–12, 12–18 years	Sample 1 = 282 Sample 2 = 1144 Sample 3 = 4810	Sample 1 = 20 Sample 2 = 8 Sample 3 = 8	5 1 = completely disagree 5 = completely agree	Dutch	Sample 1 = 0.80 Sample 2 = 0.80 Sample 3 = 0.83			β	

Note: β**:** no data present; δ: 4 = often, 3 = sometimes, 2 = rarely, and 1 = never.

**Table 4 ijerph-18-03285-t004:** Subsequent research papers investigating the reliability and validity of the CLS.

Authors	Year	Title	Participant Age	Number of participants	Number of items	Response Categories	Language of Sample	Cronbach’s Alpha	Mean (Standard Deviation)
Development Paper: Asher, Hymel and Renshaw [9]	1984	Loneliness in children	Grades 3 to 6	506	24	5 (1 = always true, 2 = true most of the time, 3 = true sometimes, 4 = hardly ever true, 5 = not true at all)	English	0.90	32.51 (11.82)
Cassidy and Asher [47]	1992	Loneliness and peer relations in young children	5–7 years	452	23	3 “Yes” “No” “Sometimes”	English	0.79	β
Goossens and Beyers [33]	2002	Comparing measures of childhood loneliness: internal consistency and confirmatory factor analysis	Grades 5–6	292	24	5 η	Dutch	0.87	33.11 (9.98)
Bagner, Storch, and Roberti [48]	2004	A factor analytic study of the loneliness and social dissatisfaction scale in a sample of african american and hispanic-american children	10–13 years	200	24	5	English	0.87*Boys =* 0.84*Girls =* 0.84	Ethnicity: African American = 29.63 (11.71) Hispanic = 33.92 (12.42) Grade: Fifth Grade = 35.42 (13.74) Sixth grade = 31.65 (11.10) Gender: Boys = 33.43 (11.92) Girls = 33.07 (12.81)
Coplan, Closson, and Arbeau[49]	2007	Gender differences in the behavioural associates of loneliness and social dissatisfaction in kindergarten.	Mean = 64.76 months δ	139	16	3	English	0.76	β
Ebesutani, Drescher, Reise, Heiden, Hight, Damon, and Young [50]	2012	The loneliness questionnaire-short version: an evaluation of reverse-worded and non-reverse-worded items via item response theory.	Grades 2–12	12722	24	5	English	Reverse-worded items = 0.73 Non-reverse worded items = 0.92	β
Ritchwood, Ebesutani, Chin, and Young [51]	2017	The loneliness questionnaire: establishing measurement invariance across ethnic groups	Grades 2–12	12344	24	5 η	English	African American sample= 0.85 Caucasian sample= 0.88	β
**‘Pure’ Loneliness papers**									
Parker and Asher[52]	1993	Friendship and friendship quality in middle childhood: links with peer group acceptance and feelings of loneliness and social dissatisfaction	Grades 3–5 θ	881	3 *Σ*	β	English	0.77	β
Ladd, Kochenderfer, and Coleman[53]	1996	Friendship quality as a predictor of young children’s early school adjustment	Average age = 5.61 years	82	5 λ	3 “no”, “sometimes”, “yes”	English	Autumn = 0.75 Spring = 0.78	β
Ladd, Kochenderfer, and Coleman [54]	1997	Classroom peer acceptance, friendship, and victimization: distinct systems that contribute uniquely to children’s school adjustment.	Average age = 5:6 years	200	5 λ	3	English	Autumn = 0.75 Spring = 0.78	β
Rotenburg, McDougall, Boulton, Vaillaincourt, Fox, and Hymel [55]	2004	Cross-sectional and longitudinal relations among peer-reported trustworthiness, social relationships, and psychological adjustment in children and early adolescents from the united kingdom and Canada	9–11 years Mean age = 9:9	*Time 1 =* 505 *Time 2 =* 475	4 Σ Additional item = “I have no one to talk to”	5	English	0.87	β

β: data not present; δ: 5 years, 3 months; θ: 7–10 years; η: see development paper response categories; Σ: Items included (1) I feel alone, (2) I feel left out, (3) I am lonely at school; λ: 3 items directly referring to loneliness, plus (4) Are you sad and alone at school? (5) Is school a lonely place for you?

## Data Availability

All data are available in the table herein.

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
