# Peer review of "A Systematic Review of the Development and Psychometric Properties of Loneliness Measures for Children and Adolescents"

_ijerph, 2021, doi:10.3390/ijerph18063285_

Round 1

Reviewer 1 Report

This is an interesting and useful article, it is carefully constructed, and the authors' reasoning is clearly and in detail explained.

However, the article suffers from several shortcomings that need to be corrected.

  • In the section describing the Literature Search Strategy and Review Process, different numbers are given than in the PRISMA diagram. The same problem occurs in the sample description in Section 3.1 Scale Overview and in the following Tables 2-4. Such inconsistencies reduce the credibility of the text.
  • In section 3.2, Quality Appraisal, lines 223 -225, the authors state that the details of the definition of good data analysis processes are described in Supplementary Table S2 (and mention the correlations of the loneliness scale and relevant related constructs), but this is not stated in S2.
  • According to heading 3.3.1. Reliability and Validity, the text should also comment on the validity of the measures, but the authors apparently did not examine the validity. This is visible from both above point 2) and from the fact that the issue of validity is not elaborated in the Literature Search Strategy (the term “validity” was used, but no alternatives enabling for inspection of concurrent, predictive or ecological validity like e.g. correlations, predictions, etc.). I, therefore, recommend that the authors use the word validity in the text more appropriately.
  • In terms of the inclusion of selected studies, the question arises as to how the authors viewed the question of the respondents’ age. For example, they took for UCLA into account also studies that did not correspond to the development paper with age. Nevertheless, if the age range of respondents widens significantly (e.g., in UCLA, the authors report on studies involving 11-18 years, in CLS 5 -13 years), construct validity is questionable. Is loneliness, as it is operationalized by the questions of the questionnaires, the same for developmentally different cohorts of respondents? I think that this issue should be included in the discussion.
  • According to the discussion of reliability values ​​in the text, it seems that the authors assume that the higher the value of Cronbach's alpha the better, and that it also applies above the value ​​ 0.9. However, this is not true, a maximum alpha value of 0.90 is widely recommended. If alpha is too high, it may suggest that some items are redundant as they are testing the same question but in a different guise.

Author Response

Please see the attachment and yellow highlighted sections in manuscript. 

Reviewer 2 Report

I consider that the article deserves a prompt publication, although it is recommended to review some elements:

a) The information contained in table 2: Relative to the sample.

It is not clear what the numbers 1,2,3,4 correspond to. It is possible to think that they are different applications, but for example in the case of Danneel, Maes, Vanhalst, Bijttebier & Goossens there are only three-wave.

Therefore, a clarification of the information contained in the table is requested by incorporating all the explanations in the table footnote.

b) Discussion

The data is not sufficiently discussed and as it is formulated it can be understood as part of the conclusion.

In particular, it is recommended to make explicit, based on the literature, as indicated in (line 314), which specific aspects of the evidence should be reviewed.

Therefore, it is recommended to complete a general review of the Discussion.

Author Response

(The authors gave the same response as above.)

Round 2

Reviewer 1 Report

The revised text met all the points of the previous review in an appropriate manner and the text is significantly improved. No other suggestions for refinement except for check for few typos.